# Bacterial Colonization Incidence before and after Indwelling Double-J Ureteral Stents

**DOI:** 10.3390/antibiotics11070850

**Published:** 2022-06-24

**Authors:** Sholpan Kaliyeva, Natalya Simokhina, Alyona Lavrinenko, Gulzira Zhussupova, Serik Zhunusov, Polina Semenikhina, Yuliya Bikbatyrova, Berik Yelmagambetov, Zhanna Myasnikova

**Affiliations:** 1Department of Clinical Pharmacology and Evidence-Based Medicine, NCJSC Karaganda Medical University, Karaganda 100000, Kazakhstan; s-kalieva@qmu.kz (S.K.); kornienko@kgmu.kz (Y.B.); myasnikovaz@qmu.kz (Z.M.); 2Scientific Research Laboratory, NCJSC Karaganda Medical University, Karaganda 100000, Kazakhstan; lavrinenko.alena@gmail.com; 3National Center for Rational Use of Medicines, Nur-Sultan 010000, Kazakhstan; 4Department of Surgery Diseases, NCJSC Karaganda Medical University, Karaganda 100000, Kazakhstan; zhunusov@qmu.kz; 5Neurology, Psychiatry and Rehabilitation Department, NCJSC Karaganda Medical University, Karaganda 100000, Kazakhstan; semenihina@qmu.kz; 6“SANAT” National Education Development Science Center, Nur-Sultan 010000, Kazakhstan; berami@mail.ru

**Keywords:** antibiotics, antibiotic susceptibility, antimicrobial resistance, antibiotic stewardship, stenting, upper urinary tract, urinary tract infections

## Abstract

The upper urinary tract stenting allows to restore the ureteral patency in various situations. However, one of the main disadvantages of stenting is bacterial contamination, which can be a source of persistent infections that hardly respond to antibiotic therapy. The aim of this study was to investigate the local spectrum of bacterial pathogens and their susceptibility to antibiotics in order to optimize antibacterial therapy after upper urinary tract stenting. A prospective observational study was conducted in which 140 urine samples were examined (70 before stenting and 70 after stenting). Bacterial growth was detected in 37 patients (52.8%) before stenting and in 43 patients (61.4%) after stenting. *E. coli* (13 (28.8%)) and *Streptococcus spp.* (8 (17.6%)) strains were more commonly detected before stenting; *P. aeruginosa* (15 (31.2%)) and *E. coli* (8 (16.6%)) were usually revealed after stenting. The proportion of *P. aeruginosa* strain*s* after stenting grew from 4.4% up to 31.2%. *E. coli* strains were resistant to ampicillin (92.3% before and 100% after stenting). Three strains of *E. coli* (23.1%) and six strains of *P. aeruginosa* (40%) were multidrug-resistant. Determination of the bacterial sensitivity to antibiotics and identification of antibiotic-resistant forms of bacteria is a factor in reducing the risk of complications and optimizing antibiotic therapy during the upper urinary tract stenting.

## 1. Introduction

Urinary tract stenting is a minimally invasive surgery that aims to restore the ureteral patency and normalize the outflow of urine from the kidney. Upper urinary tract stenting is an integral part of many urological surgeries [1,2,3]. The advantages of stenting include its low invasiveness, the absence of external drainage in the patient, and the increase in the patient’s quality of life. However, in addition to the obvious advantages, there are also disadvantages to the internal drainage surgeries of the upper urinary tract. One of the main problems of long-term stenting is the high probability of colonization of the stent by bacteria with the formation of antibiotic-resistant bacterial associations, which in some cases call into question the effectiveness of modern antibacterial therapy [4,5,6].

According to the US National Health Safety Network (NHSN), the prevalence of stent-associated complications is 3.1–7.5 cases per 1000 catheter days [7,8]. The risk of bacteriuria and stent colonization can significantly increase due to the continuous presence of the stent, concomitant diseases such as diabetes mellitus, chronic renal failure, diabetic nephropathy, recent or recurrent urinary tract infections, and pregnancy [9,10,11]. The bacteria most commonly detected during stent colonization are *Escherichia coli (E. coli)*, *Proteus mirabilis (P. mirabilis)*, *Pseudomonas aeruginosa (P. aeruginosa)*, *Enterococcus faecium (E. faecium)* [12,13,14]. Many strains of microorganisms excreted in patients after ureteral stent implantation are characterized by drug resistance, including multidrug resistance, which causes significant difficulties in the treatment of the aforementioned infections in hospitalized patients, especially in the postoperative period [15].

Considering that the presence of antibiotic-resistant forms of bacteria after urinary tract stenting can lead to the development of multiple complications in urological patients, the urgent task and purpose of this research were to investigate the local spectrum of bacterial pathogens and their susceptibility in order to optimize antibiotic therapy after upper urinary tract stenting.

## 2. Results

### 2.1. General Characteristics of the Study

The study was conducted between January and March 2020. The age of patients ranged from 19 to 79 years, with a mean of 49 years (95% CI: 45.2–52.8). A total of 53% of participants (*n* = 37) who underwent stenting were men and 47% (*n* = 33) were women.

### 2.2. Results of the Study on the Spectrum of the Most Important Bacterial Pathogens in Urinary Tract Infections before and after the Upper Urinary Tract Stenting

According to the results of a microbiological study, the growth of microorganisms was detected in 37 patients (52.8%) before upper urinary tract stenting, while 33 patients (47.2%) were abacterial. After the upper urinary tract stenting, the growth of microorganisms was detected in 43 patients (61.4%), while growth was absent in 27 patients (38.6%). The increase in the frequency of detection of microorganisms after stenting was not statistically significant (*p* = 0.289).

The etiological structure of pathogens obtained from patients prior to upper urinary tract stenting included microorganisms of the *Enterobacterales* order in 53% of cases [*n* = 24], *Enterococcus spp.* in 13.2% [*n* = 6], *Streptococcus spp.* in 11.1% [*n* = 5], *Staphylococcus spp* in 11.1% [*n* = 5], *Burkholderia* (*B. cenocepacia*) in 4.4% [*n* = 2], *Pseudomonas spp* (*P. aeruginosa*) in 4.4% [*n* = 2], *Acinetobacter* (*A. baumannii*) in 2.2% [*n* = 1].

The etiological structure of the pathogens obtained from patients after upper urinary tract stenting contained the following microorganisms: *Pseudomonas spp.* in 33.3% [*n* = 16], *Enterobacterales* in 31.3% [*n* = 15], *Staphylococcus spp.* in 12.5% [*n* = 6], *Enterococcus spp.* in 12.5% [*n* = 6], *Burkholderia* (*B. cenocepacia*) in 6.2% [*n* = 3]*, Streptococcus spp.* (*S. mitis*) in 2.1% [*n* = 1], and *Acinetobacter* (*A. baumannii*) in 2.1% [*n* = 1] (Table 1).

When comparing the nature of microflora in patients before and after upper urinary tract stenting, it was found that out of 37 patients in whom bacterial growth was detected before stenting, after stenting, the pathogen changed in 18 patients (48.6%), the bacterial flora remained the same in 6 patients (16.2%), and there was no bacterial growth in 13 patients (35.1%). Of the 33 patients in whom no growth of microorganisms was detected before stenting, after stenting, bacteria were detected in 19 patients (57.6%) and still absent in 14 patients (42.4%). A comparison of the frequency of changes in the microbial flora using the McNemar’s test showed that the values were statistically significantly different in the patients before and after stenting (*p* < 0.001) (Figure 1).

The following changes in the number of strains were revealed for the individual microorganisms: *E. coli* was detected in 13 patients (18.6%) before stenting; however, after stenting, this pathogen was no longer observed in 12 of these patients (17.1%). The decrease in the frequency of *E. coli* detection was statistically significant (*p* = 0.015).

*S. anginosus* was detected in four patients (5.7%) before stenting, while after stenting, this pathogen was completely absent in these patients. The decrease in the frequency of detection of *S. anginosus* is also statistically significant (*p* = 0.046).

*P. aeruginosa* was detected in 13 patients (18.6%) after stenting in whom this pathogen had not previously been present. Only in two patients (2.8%) *P.aeruginosa* strains were detected both before and after stenting.

In 19 patients (27.1%) who initially had no microflora growth, various strains of *P. aeruginosa*, *E. coli, S. haemolyticus,* and *E. faecium* were isolated after stenting.

### 2.3. Determination of Sensitivity to Antibacterial Drugs in Microorganisms Isolated before Upper Urinary Tract Stenting

The results of in vitro antibiotic susceptibility testing of *E. coli* strains isolated before stenting showed that most microorganisms were susceptible to ampicillin/sulbactam, piperacillin/tazobactam, cephalosporins, carbapenems, aminoglycosides, and fluoroquinolones. *E. coli* strains were ampicillin-resistant in 92.3% of cases. A total of three strains (23.1%) of *E. coli* were multidrug-resistant (MDR) (i.e., were resistant to at least one antimicrobial agent in three or more chemical classes of antibiotics) [16].

Data obtained during the study revealed that most of the isolated *S. anginosus* strains were sensitive to amoxicillin/clavulanic acid, vancomycin, lincomycin, tetracycline, linezolid, and fluoroquinolones.

All isolated *E. faecium* strains in our study were characterized by high sensitivity to amoxicillin/clavulanic acid, vancomycin, and ofloxacin and were resistant to ciprofloxacin.

Meanwhile, all strains of *K. ozaenae* isolated during the study showed sensitivity to ampicillin/sulbactam, colistin, cephalosporins, carbapenems, aminoglycosides, and fluoroquinolones. Most strains were moderately sensitive to ceftriaxone and resistant to piperacillin/tazobactam, and cefuroxime. The antibiotic sensitivity of the main strains of microorganisms isolated before stenting is presented in Table 2.

Thus, of the 23 antimicrobial drugs used in the treatment of patients prior to upper urinary tract stenting, only 6 belong to the access group according to the WHO AWARE classification, and two belong to the reserve group.

### 2.4. Determination of Sensitivity to Antibacterial Drugs in Microorganisms Isolated after Upper Urinary Tract Stenting

The results of in vitro antibiotic susceptibility testing of *P. aeruginosa* strains isolated after stenting showed that the strains were highly sensitive to colistin, polymyxin B, and fosfomycin. The strains showed resistance to ticarcillin/clavulanic acid, ceftazidime, cefepime, ciprofloxacin carbapenems, and aminoglycosides. Six *P. aeruginosa* strains (40%) were multidrug-resistant.

The *E. coli* strains isolated after stenting showed sensitivity to III and IV generation of cephalosporins, carbapenems, aminoglycosides, and fluoroquinolones. All isolated *E. coli* strains were ampicillin-resistant. A total of two strains of *E. coli* showed resistance to all drugs of the cephalosporin group.

All isolated strains of *E. faecalis* were sensitive to ampicillin, imipenem, vancomycin, and linezolid and resistant to drugs from the fluoroquinolone group.

The isolated *S. haemolyticus* strains were sensitive to tetracycline, linezolid, vancomycin, and teicoplanin; 50% of the strains were resistant to ciprofloxacin, azithromycin, and gentamicin.

*B. cenocepacia* in our study were sensitive to piperacillin, ceftazidime, cefepime, aztreonam, meropenem, and tobramycin; 66.6% of the strains showed resistance to gentamicin and ciprofloxacin. The antibiotic sensitivity of the most important strains of microorganisms isolated after stenting is presented in Table 3.

It stands to mention that the application pattern for antimicrobial drugs used in the treatment of patients after upper urinary tract stenting is different from that before stenting. Thus, of the 24 antimicrobial drugs, 4 belong to the access group according to the WHO AWARE classification, 4 belong to the reserve group, and the TCC group is not delineated in the WHO AWARE classification.

## 3. Discussion

The most common microorganism before stenting in our study was *E. coli*, which is consistent with the data from other researchers who report that detection of *E. coli* colonizing stents is one of their most frequent findings [17,18,19,20].

Of the 13 patients (18.6%) who had *E. coli* before stenting, 12 patients (17.1%) after stenting lacked *E. coli* growth, which can be explained by the rational choice of antibiotic therapy. The rationality of antibiotic therapy can also be evidenced by the fact that in four patients (5.7%), who demonstrated *S. anginosus* growth before stenting, this pathogen was completely absent after stenting and antibiotic therapy.

After upper urinary tract stenting, the most common microorganism in our study was *P. aeruginosa*, which was isolated in one-third of all patients. *P. aeruginosa* is one of the leading nosocomial pathogens of urinary tract infections (more often, as in our case, catheter-associated), which has been confirmed by data from multiple other studies [12,20]. *P. aeruginosa* infections rank third among nosocomially acquired urinary tract infections associated with catheters [21,22,23]. *P. aeruginosa* strains were detected in two patients (2.8%) both before and after stenting; this fact might indicate an irrational selection of antibiotic therapy. The proportion of *P. aeruginosa* strains among all strains in our study grew from 4.4% before stenting up to 31.2% after stenting. We assume that the increase in the proportion of *P. aeruginosa* strains after stenting may have happened due to both lack of antibiotic prophylaxis and prolonged antimicrobial therapy, as well as non-compliance with the principles of anti-infection protection leading to nosocomial infections with *P. aeruginosa* strains. According to S. J. Childs, urological instruments are the main cause of urinary tract infections during hospitalization in 75–80% of cases [24]. The identification of *P. aeruginosa* leads to considerable difficulties in choosing adequate empirical therapy, especially for multidrug-resistant strains, resulting in an increase in mortality and prolonged hospitalization time.

Despite the long duration of antibacterial therapy, we found a change in pathogens in 18 patients after stenting, which may indicate that the spectrum of prescribed antibiotics did not include identified pathogens. In addition, even though urine samples for the determination of antibiotic sensitivity were collected from all patients, antibacterial therapy was changed only for 11 of them (15.7%), indicating that urine culture data were not always taken into account during the treatment and antibiotics were prescribed according to a conventional template.

Another important nosocomial pathogen is *B. cenocepacia*, which was isolated in two (4.4%) patients before stenting and three patients (6.2%) after stenting in our study. Most commonly, this pathogen causes infections of the respiratory system in patients with cystic fibrosis [25,26], but cases of the urinary tract infections caused by *B. cenocepacia* have also been described before [26,27,28]. Given that *B. cenocepacia*, especially multidrug-resistant strains, can cause increased mortality, detection of this pathogen in urological patients is a serious threat. The rest of the pathogenic strains isolated in the course of our study are also among the commonly detected microorganisms in urinary system infections, according to other researchers [10,12,17,18,29,30,31,32,33]. Strains of *P. aeruginosa*, *E. coli*, *S. haemolyticus,* and *E. faecium* were detected after stenting in 19 patients (27.1%) who were initially abacterial, which may indicate that upper urinary tract stenting is a risk factor for the occurrence of stent-associated urinary tract infections.

In our study, all patients initially received empirical antibiotic therapy, which was later adjusted depending on the uroculture results. However, the study involved both patients with and without signs of infection, and the second group could only receive a prophylactic dose of antibiotics according to clinical recommendations [34,35,36]. Nevertheless, none of the patients included in the study received prior antibiotic prophylaxis, which is irrational and not in line with the international clinical recommendations [34,35,36]. Prophylactic administration of antibiotics could have helped to reduce the risk of infection in patients in the postoperative period.

Most of the *E. coli* strains were resistant to ampicillin both before and after stenting, and some of the strains showed multi-resistance. However, most of the strains showed sensitivity to antibacterial drugs commonly used in urological practice, which led to a decrease in the number of *E. coli* strains identified after treatment. *P. aeruginosa* strains demonstrated multidrug resistance in 40% of cases, which was a significant problem in the treatment of urinary tract infections, especially catheter-associated ones, and potentially led to unfavorable outcomes of the disease. Our data are partially consistent with those of other researchers [12,20,37]; however, there are differences in the spectrum of antibiotic resistance that can be explained by the local characteristics of bacterial antibiotic susceptibility.

The manifestation of increasing antibiotic resistance is also confirmed by a change in antimicrobial drugs used after upper urinary tract stenting toward the prescription of reserve antibiotics (with 2 reserve antibiotics before stenting and 4 after stenting), which undoubtedly raises concern.

The data obtained in our study formed the basis for the creation of recommendations to optimize the prophylactic and therapeutic use of antibiotics in upper urinary tract stenting, which were incorporated into the work of the urological department of the regional hospital. Although local data on the most common microorganisms and their sensitivity to antibiotics are specific to an individual hospital, these data can still influence antibiotic use policies in the region and country.

There were several limitations to be aware of when interpreting our results. First, we used a limited sample size, reflecting the fact that the recruitment of patients occurred at the beginning of the COVID-19 pandemic when hospitalization rates for diseases other than COVID-19 (including urologic diseases) diminished severely. Second, the data we obtained on antibiotic resistance of the isolated strains were local and characteristic of a particular region, which limits the possibility of generalizing our results. Third, all patients received empiric antibiotic therapy for at least 48–72 h, which could have influenced the results of the study.

## 4. Materials and Methods

### 4.1. Ethical Approval

The study followed the principles of scientific ethics. All research procedures were conducted in accordance with the ethical standards approved by the legal regulations of the Republic of Kazakhstan in the field of clinical trials regulation and the Declaration of Helsinki of the World Medical Association on the Ethical Principles of Medical Research involving Human Subjects. All study participants were informed about the aims of the study and signed written informed consent. Patients had the right to choose whether or not to participate in the study and were included in the study only voluntarily. All participants’ data were entered into the database, and their identifiers were encrypted. All information obtained during the study will be used for scientific purposes only and is strictly confidential. Ethical approval for the study was obtained from the Bioethics Committee of the Karaganda Medical University (Protocol #27 dated 23 December 2019).

### 4.2. Study Design

The study was conducted between January and March 2020. Sampling was carried out in the urological department of the regional hospital. Microbiological research was performed in the shared laboratory of the Karaganda Medical University. A total of 70 patients who had undergone upper urinary tract stenting participated in the study. The material for the study included 140 urine samples collected from patients before and after upper urinary tract stenting. Informed consent to participate in the study was obtained from all patients.

Inclusion criteria: male and female patients over 18 years of age who were scheduled for upper urinary tract stenting and who had given informed consent to participate in the study, both with and without signs of infection.

Exclusion criteria: patients under 18 years of age; pregnant women; patients with oncologic diseases, chronic liver and kidney diseases, diabetes mellitus in decompensation stage; patients having taken antibacterial drugs in the last 3 months.

All patients received antibacterial therapy upon admission to the hospital. In all cases, antibacterial drugs were initially prescribed empirically, and after obtaining the results of a bacteriological study (48–72 h later), antibacterial therapy was adjusted in accordance with the sensitivity of the microflora to antibiotics. Antibacterial therapy was prescribed in accordance with the clinical protocols for diagnosis and treatment in the Republic of Kazakhstan [38,39] and lasted 5–10 days. The most commonly prescribed empiric antibiotic therapy included cephalosporins (cefuroxime, ceftazidime, ceftriaxone) and fluoroquinolones (ofloxacin, levofloxacin, ciprofloxacin). The results of urine culture sensitivity tests promoted the antibiotic therapy changes only in 11 patients (15.7%).

Urine samples were collected from patients before and after urinary tract stenting. Urine samples before stenting were collected from patients on the day of hospital admission; surgery was usually performed on the day of admission or the next day. Post-stenting urine samples were collected on the day of discharge from the hospital (usually 5–10 days after surgery). After receiving instructions for urine collection, each patient was given a container for urine. No later than 2 h after collection, the biological material was delivered to the shared laboratory of the Karaganda Medical University Research Center in accordance with sanitary standards.

All investigated isolates were identified as species by using matrix-assisted laser desorption/ionization—time-of-flight mass spectrometry (MALDI-TOF MS) using the Microflex LT system and MALDI Biotyper Compass 4.1.80 software (Bruker Daltonics, Bremen, Germany) [40]. Identification was considered successful at the species level with a high degree of confidence if the score exceeded 2.0; if the score was between 2.0 and 1.7, identification was considered successful at the genus level with sufficient confidence [40].

The susceptibility of bacterial isolates to antibacterial drugs was investigated by the disk diffusion method to the following antibiotics: oxacillin (1 µg), cefoxitin (30 µg), ampicillin (10 µg), cefepime (30 µg), imipenem (10 µg), meropenem (10 µg), amikacin (30 µg), kanamycin (30 µg), streptomycin (10 µg), gentamycin (10 µg), netilmicin (30 µg), tobramycin (30 µg), nalidixic acid (30 µg), ciprofloxacin (5 µg), levofloxacin (5 µg), linezolid (30 µg), clindamycin (2 µg), rifampicin (5 µg), azithromycin (15 µg), fusidic acid (10 µg), and tetracycline (30 µg). The analysis of the results was performed following the recommendations of CLSI 2019 [41]. Determination of bacterial isolate susceptibility to colistin was carried out using the broth microdilution method according to the EUCAST guidelines (2019) [42,43].

Internal quality control was carried out on the control strains: *Escherichia coli* ATCC 25,922 (ESBL and AmpC−), *Pseudomonas aeruginosa* ATCC 27853, *Klebsiella pneumoniae* WHO-3 (ESBL +), *Enterobacter cloacae* WHO-238 (AmpC + and ESBL−).

The analysis of susceptibility to antibacterial drugs was performed by calculating the 95% confidence interval using the WHONET 5.6 program.

### 4.3. Statistics

Statistical analysis of the research results was carried out using the STATISTICA 13.5.0 application package and the Excel spreadsheet from the Microsoft Office 2014 software package. The reliability criterion *p* < 0.05 was considered statistically significant for all indicators. In order to determine statistically significant differences in the analysis of qualitative indicators in two dependent groups, contingency tables with McNemar’s test were used [44].

## 5. Conclusions

Our study demonstrated that *E. coli* was the most frequently detected pathogen before stenting and *P. aeruginosa*—after stenting; 19 patients (27.1%) who were initially abacterial demonstrated growth of microflora after stenting. These differences may indicate that upper urinary tract stenting is a risk factor for stent-associated urinary tract infections. In addition, the increase in the number of *P. aeruginosa* strains after stenting could be due to insufficient adherence to the principles of anti-infective protection. Lack of antibiotic prophylaxis may also contribute to the appearance of bacteria in postoperative patients. A total of three strains of *E. coli* (23.1%) and six strains of *P. aeruginosa* (40%) were multidrug-resistant, explaining the increased need for reserve antibiotics after stenting.

## Figures and Tables

**Figure 1 antibiotics-11-00850-f001:**
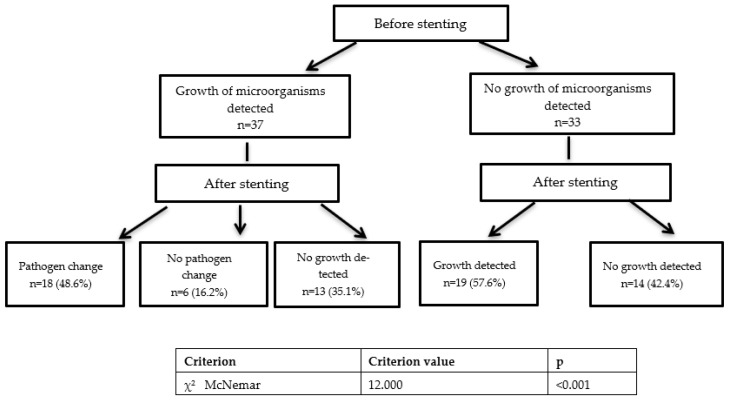
Changes in microflora after the upper urinary tract stenting.

**Table 1 antibiotics-11-00850-t001:** Microorganisms detected in urine samples before and after upper urinary tract stenting.

Microorganism	Before Stenting *n* (%)	After Stenting *n* (%)
** *Enterobacterales* **	24 (53%)	15 (31.3%)
*E.coli*	13 (28.8%)	8 (16.6%)
*K. ozaenae*	4 (8.8%)	1 (2.1%)
*K. oxytoca*	3 (6.6%)	2 (4.2%)
*K. pneumoniae*	2 (4.4%)	1 (2.1%)
*E. cloacae*	1 (2.2%)	-
*E. aerogenes*	1 (2.2%)	2 (4.2%)
*P. mirabilis*	-	1 (2.1%)
** *Pseudomonas spp.* **	2 (4.4%)	16 (33.3%)
*P. aeruginosa*	2 (4.4%)	15 (31.2%)
*P. fluorescens*	-	1 (2.1%)
** *Streptococcus spp.* **	5 (11.1%)	1 (2.1%)
*S. anginosus*	4 (8.8%)	-
*S. mitis*	1 (2.2%)	1 (2.1%)
** *Enterococcus spp.* **	6 (13.2%)	6 (12.6%)
*E. faecium*	4 (8.8%)	2 (4.2%)
*E. faecalis*	2 (6.6%)	4 (8.4%)
** *Staphylococcus spp.* **	5 (11.1%)	6 (12.5%)
*S. haemolyticus*	3 (6.6%)	4 (4.8%)
*S. hominis*	2 (4.4%)	1 (2.1%)
*S. epidermidis*	-	1 (2.1%)
** *Burkholderia* **	2 (4.4%)	3 (6.2%)
*B. cenocepacia*	2 (4.4%)	3 (6.2%)
** *Acinetobacter* **	1 (2.2%)	1 (2.1%)
*A. baumannii*	1 (2.2%)	1 (2.1%)
Total number	45	48

**Table 2 antibiotics-11-00850-t002:** Antibiotic sensitivity of the main strains of microorganisms isolated before stenting.

Antibiotic	*E. coli*, *n* (%)	*S. anginosus*, *n* (%)	*E. faecium*, *n* (%)	*K. ozaenae*, *n* (%)	AWARE
AMP	1 (7.7%)				access
AMS	13 (100%)			4 (100%)	access
AMC		3 (75%)	4 (100%)		access
PTZ	9 (69%)			2 (50%)	watch
CAZ	12 (92.3%)			4 (100%)	watch
CTX	12 (92.3%)			4 (100%)	watch
CXM	11 (84.6%)			2 (50%)	watch
CRO	11 (84.6%)			3 (75%)	watch
FEP	12 (92.3%)			4 (100%)	watch
IPM	13 (100%)			4 (100%)	watch
MER	13 (100%)			4 (100%)	watch
TOB	13 (100%)			4 (100%)	watch
GEN	10 (76.9%)			4 (100%)	access
AMI	12 (92.3%)			4 (100%)	access
OFX	13 (100%)	4 (100%)	4 (100%)		watch
LVX	12 (92.3%)	4 (100%)		4 (100%)	watch
CIP	8 (61.5%)	4 (100%)	0 (0%)	4 (100%)	watch
NOR		4 (100%)			watch
VAN		4 (100%)	4 (100%)		watch
LIN		4 (100%)			watch
LNZ		4 (100%)			reserve
TCY		3 (75%)			access
COL				4 (100%)	reserve

The number of susceptible strains of bacteria (percentage), AMP—ampicillin, AMS—ampicillin/sulbactam, AMC—amoxicillin/clavulanic acid, PTZ—piperacillin/tazobactam, CAZ—ceftazidime, CTX—cefotaxime, CRO—ceftriaxone, CXM—cefuroxime, FEP—cefepime, IPM—imipenem, MER—meropenem, TOB—tobramycin, GEN—gentamicin, AMI—amikacin, OFX—ofloxacin, LVX—levofloxacin, CIP—ciprofloxacin, NOR—norfloxacin, VAN—vancomycin, LIN—lincomycin, LNZ—linezolid, TCY—tetracycline, COL—colistin.

**Table 3 antibiotics-11-00850-t003:** Antibiotic sensitivity of the main strains of microorganisms isolated after stenting.

Antibiotic	*P. aeruginosa* *n* (%)	*E. coli* *n* (%)	*E. faecalis* *n* (%)	*S. haemolyticus*, *n* (%)	*B. cenocepacia*, *n* (%)	AWARE
AMP		0 (0%)	4 (100%)			access
PIP					3 (100%)	access
TCC	0 (0%)					access
CAZ	2 (13.3%)	6 (75%)			3 (100%)	watch
CTX		5 (62.5%)				watch
CRO		6 (75%)				watch
FEP	3 (20%)	7 (87.5%)			3 (100%)	watch
IPM	6 (40%)	8 (100%)	4 (100%)			watch
MER	4 (26.7%)	8 (100%)			3 (100%)	watch
ATM					3 (100%)	watch
TOB	8 (53.3%)	8 (100%)			3 (100%)	watch
GEN		8 (100%)		2 (50%)	1 (33.3%)	watch
AMI	9 (60%)	8 (100%)				access
OFX		8 (100%)				access
LVX		8 (100%)	0 (0%)			watch
CIP	4 (26.7%)		0 (0%)	2 (50%)	1 (33.3%)	watch
VAN			4 (100%)	4 (100%)		watch
TEIC				4 (100%)		watch
LNZ			4 (100%)	4 (100%)		watch
TCY				4 (100%)		watch
COL	9 (100%)					reserve
POL	15 (100%)					access
FOS	6 (100%)					reserve
AZM				2 (50%)		

The number of susceptible strains of bacteria (percentage), AMP—ampicillin, PIP—piperacillin, TCC—ticarcillin/clavulanic acid, AMS—ampicillin/sulbactam, AMC—amoxicillin/clavulanic acid, PTZ –piperacillin/tazobactam, CAZ—ceftazidime, CTX—cefotaxime, CRO—ceftriaxone, CXM—cefuroxime, FEP—cefepime, IPM—imipenem, MER—meropenem, ATM—aztreonam, TOB—tobramycin, GEN—gentamicin, AMI—amikacin, OFX—ofloxacin, LVX—levofloxacin, CIP—ciprofloxacin, NOR—norfloxacin, VAN—vancomycin, TEIC—teicoplanin, LIN—lincomycin, LNZ—linezolid, TCY—tetracycline, COL—colistin, POL—polymixin B, FOS—fosfomycin, AZM—azithromycin.

## Data Availability

The data presented in this study are available on request from the corresponding author.

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
