# Peer review of "Bacterial Colonization Incidence before and after Indwelling Double-J Ureteral Stents"

_antibiotics, 2022, doi:10.3390/antibiotics11070850_

Round 1
Reviewer 1 Report
Thanks for your work, i found the paper interesting from a microbiological standpoint, however, I have some questions for you.
In the first instance, your goal was to study the spectrum of pathogens encountered in your center before and after ureteral stent placement. However, with regard to the study design I found the following to be lacking: 1) the study period, 2) how patients were chosen to enter the study (were all eligible patients included according to the inclusion/exclusion criteria or were only patients whose records were available?) 3) the authors mention urine cultures performed before and after surgery, but lack data on when they were collected (performing a urine culture many days before surgery and many days after could have had an impact on the study).
On the other hand, regarding the results: if your goal was to assess ecology you probably need to add something to the analysis. To say that the etiology has changed is quite crude, it would have been useful to analyze the reasons for the change (e.g.: the increase in Pseudomonas, which you empirically explain by the lack of adherence to prophylaxis measures, could have other causes?, was the type of intervention always performed in the same way?, was the criterion of urgency or election considered? comorbidities?).
Another factor that I recommend investigating is antibiotic therapy, in how many patients was this done? Could this have played a role in the selection of different pathogens?
Author Response
Response to Reviewer 1 Comments
Dear Reviewer!
Thank you very much for your comments and recommendations!
Please find out our answers for your comments below.
Point 1: I found the following to be lacking: the study period.
Response 1: the period of the study (from January to March 2020) was mentioned in the "Results" section, we will duplicate this information by adding it to the "Materials and Methods" section.
Point 2: I found the following to be lacking: how patients were chosen to enter the study (were all eligible patients included according to the inclusion/exclusion criteria or were only patients whose records were available?
Response 2: In the "Materials and Methods" section, we presented the inclusion and exclusion criteria for patients:
“Inclusion criteria: male and female patients over the age of 18 years who were planned to undergo upper urinary tract stenting and gave informed consent to participate in the study both with signs of infection and without them.”
“Exclusion criteria: patients under 18 years of age; pregnant women; patients with on-cological diseases, chronic liver and kidney diseases, diabetes mellitus in the stage of decompensation; patients having taken antibacterial drugs in the last 3 months.”
The study was prospective and we included all eligible patients, as we were able to perform all needed tests and assess their records.
Point 3: The authors mention urine cultures performed before and after surgery, but lack data on when they were collected (performing a urine culture many days before surgery and many days after could have had an impact on the study).
Response 3: Urine samples before stenting were collected from the patients on the day of admission to the hospital; surgery was usually performed on the day of admission or the next day. Urine samples after stenting were collected on the day of discharge from the hospital (usually 5-10 days after surgery).
This information has been added to the text of the article.
Point 4: It would have been useful to analyze the reasons for the change (e.g.: the increase in Pseudomonas, which you empirically explain by the lack of adherence to prophylaxis measures, could have other causes?, was the type of intervention always performed in the same way?, was the criterion of urgency or election considered? comorbidities?).
Response 4: The type of intervention for all patients was the same - transurethral removal of obstruction from the ureter and/or renal pelvis. In the majority of our patients, surgery was performed urgently.
Our study did not include patients with any concomitant diseases in the stage of decompensation. Thus, we did not analyze the effect of concomitant diseases on the microflora changes, since this was outside the objectives of our study. Taking into account the data available to us, we hypothesized that the main reason for the detection of P.aeruginosa strains in patients after stenting could probably be non-compliance with the principles of anti-infective protection.
Point 5: Another factor that I recommend investigating is antibiotic therapy, in how many patients was this done? Could this have played a role in the selection of different pathogens?
Response 5: All patients received antibacterial therapy. In all cases, initially antibacterial drugs were prescribed empirically, after receiving the results of a bacteriological study, antibacterial therapy was adjusted in accordance with the sensitivity of the microflora to antibiotics.
Information about the changes in the number of individual microorganism strains and their alleged relationship with antibiotic therapy is included in the "Results" and "Discussion" sections.
Reviewer 2 Report
Dear Authors,
The manuscript submitted by Kaliyeva et al. presents an interesting idea, but the population included in the study is small. Infection and encrustation of double-J stent frequently occur because of its direct contact with urine. Furthermore, internal ureteral stents also offer an ideal surface for bacterial colonization and biofilm formation.
Unfortunately, the sample population included in the study is too small to extract pertinent conclusions. The results had 37 uropathogens before stenting and 43 after stenting.
Please include the limitations of the study.
Please specify aspects regarding signs of urinary tract infection in the inclusion or exclusion criteria. The maximum period of stenting?
l. 255, please specify clearly when the urine samples were collected
The Conclusions section needs to be revised according to the study results, not presenting some general aspects.
Author Response
Response to Reviewer 2 Comments
Dear Reviewer!
Thank you very much for your comments and recommendations!
Please find out our answers for your comments below.
Point 1: Please include the limitations of the study.
Response 1: There were several limitations to be aware of when interpreting our results. Firstly, we used a limited sample size, explained by the fact that the recruitment of patients was carried out during the beginning of the COVID-19 pandemic, when hospitalization rates for other diseases except COVID-19 (including urological conditions and diseases) diminished severely. Secondly, the data we obtained on the antibiotic resistance of the isolated strains are local, characteristic of a particular region, which limits the possibility to generalize our results.
This information has been added to the text of the article.
Point 2: Please specify aspects regarding signs of urinary tract infection in the inclusion or exclusion criteria. The maximum period of stenting?
Response 2: The duration of stenting from the moment of stent placement to urinalysis averaged from 5 to 10 days, the minimum period was 4 days, and the maximum period was 14 days.
The study included both patients with signs of infection and patients without signs of infection.
Point 3: l. 255, please specify clearly when the urine samples were collected
Response 3: Urine samples before stenting were collected from the patients on the day of admission to the hospital; surgery was usually performed on the day of admission or the next day. Urine samples after stenting were collected on the day of discharge from the hospital (usually 5-10 days after surgery).
This information has been added to the text of the article.
Point 4: The Conclusions section needs to be revised according to the study results, not presenting some general aspects.
Response 4:
The Conclusions section revised in accordance with the study results specifically.
Round 2
Reviewer 1 Report
Thank you for your kind response
Author Response
Thank you very much for your contribution to our manuscript. Your valuable advice has helped us to take a fresh look at some aspects of our research and try to correct our manuscript in line with your recommendations.
Reviewer 2 Report
Dear Authors,
You should try to address the suggestions from my previous review and not just mention - the information has been added.
All the patients received empirically antibiotical therapy for at least 48 h, and then according to the uroculture? According to which guidelines, they received antibiotics for such a long period.
Round 3
Reviewer 2 Report
Dear Author,
The manuscript presents some improvements but still needs some aspects to be clarified. The authors should provide a manuscript with track changes. It isn't easy to interpret the results because the international guidelines were not applied.
Strangely, the period of antibiotics administered is long, but you discovered 18 pathogen changes in your sample population. This aspect should be addressed in the discussion section.
l. 71-72 should be deleted.
l. 144, please include the definitions of MDR
l. 234-236 needs references
l. 253, please include in the limitations that all patients received empirical antibiotic therapy for at least 48-72 hours.
l.285-290, please specify the antibiotics prescribed empirically and the number of patients in which the antibiotic therapy was changed according to the urine culture results.
The Conclusion section needs to be revised. It should be more concise.
Author Response
Dear Reviewer!
Thank you very much for your comments and recommendations!
Please find out our answers for your comments below.
Point 1: Strangely, the period of antibiotics administered is long, but you discovered 18 pathogen changes in your sample population. This aspect should be addressed in the discussion section.
Response 1:
Despite the long duration of antibacterial therapy, we found a change in pathogens in 18 patients after stenting, which may indicate that the spectrum of prescribed antibiotics did not include identified pathogens. This information was added to the «Discussion» section.
Point 2: l. 71-72 should be deleted.
Response 2: l. 71-72 were deleted.
Point 3: l. 144, please include the definitions of MDR.
Response 3: The definition for «Multiple drug resistance» was added to the article.
Point 4: l. 234-236 needs references.
Response 4: References were added.
Point 5: l. 253, please include in the limitations that all patients received empirical antibiotic therapy for at least 48-72 hours.
Response 5: The information that all patients received empirical antibiotic therapy for at least 48-72 hours is now mentioned in the limitation of our study.
Point 6: l.285-290, please specify the antibiotics prescribed empirically and the number of patients in which the antibiotic therapy was changed according to the urine culture results.
Response 6: The most commonly prescribed empiric antibiotic therapy included cephalosporins (cefuroxime, ceftazidime, ceftriaxone) and fluoroquinolones (ofloxacin, levofloxacin, ciprofloxacin). The results of urine culture sensitivity tests promoted the antibiotic therapy changes only in 11 patients (15.7%).
Point 7: The Conclusion section needs to be revised. It should be more concise.
Response 7: The Conclusion section was redacted and made more concise.